# Impact and Return on Investment of the Take Kare Safe Space Program—A Harm Reduction Strategy Implemented in Sydney, Australia

**DOI:** 10.3390/ijerph182212111

**Published:** 2021-11-18

**Authors:** Christopher M. Doran, Phillip Wadds, Anthony Shakeshaft, Dam Anh Tran

**Affiliations:** 1Cluster for Resilience and Well-Being, Centre for Indigenous Health and Equity Research, Central Queensland University, Level 4, 160 Ann Street, Brisbane, QLD 4000, Australia; 2School of Social Sciences, University of New South Wales, Randwick, NSW 2052, Australia; p.wadds@unsw.edu.au; 3National Drug and Alcohol Research Centre, University of New South Wales, Randwick, NSW 2052, Australia; a.shakeshaft@unsw.edu.au (A.S.); dam.tran@unsw.edu.au (D.A.T.)

**Keywords:** harm reduction, cost–benefit analysis, return on investment, crime, violence, alcohol

## Abstract

Safe spaces are increasingly utilized to reduce alcohol-related harm, violence, crime and improve public safety in nightlife settings. This study aimed to determine the impact and return on investment of the Take Kare Safe Space (TKSS) program—a harm reduction program implemented to address alcohol-related violence and disorder in three locations in Sydney’s night-time economy between 2014 and 2019. TKSS ambassadors provided support at static safe spaces and patrolled designated nightlife precincts to provide practical assistance to vulnerable and intoxicated people. Ambassadors recorded information relating to these interactions including ‘client’ age, gender, perceived level of intoxication, time and length of engagement with the program. Costs of program implementation and benefits of major incidents averted were obtained to allow calculation of return on investment. From December 2014 to April 2019, 66,455 people were supported by TKSS ambassadors. Most users were male (62%) and aged 18–25 years (66%). Of 3633 interventions by ambassadors, serious risk of harm was averted in 735 cases (20%). The program’s return on investment is estimated at 2.67, suggesting that a $1 investment results in $2.67 in benefits. Safe Spaces are extensively utilized, particularly by young males with high levels of intoxication, and represent a positive return on investment. Despite the growth of such services, there remains a notable absence of rigorous, independent evaluation regarding the outcomes and/or social benefit of safe space programs. From a policy perspective, there is a need for more high-quality economic evaluations to better inform decisions about competing uses of limited resources.

## 1. Introduction

Alcohol use has a significant negative social and public health impact [1]. In Australia, alcohol is the sixth highest risk factor contributing to the burden of disease in Australia [2] and was responsible for 4.5% of the total burden of disease and injury in 2015 [2]. The impacts of alcohol misuse, either alone or in combination with other illicit substances, can be serious and far reaching. Short- and long-term harms include traffic accidents, injuries, aggressive and anti-social behaviour, physical and sexual violence, increased risk of chronic disease and cancer, and premature mortality [3,4,5]. These harms, to both the drinker and others, often play out in sites of night-leisure where high-risk alcohol consumption often occurs [6]. In Australia, data suggest that alcohol is a contributing factor in approximately half of all non-domestic assaults that occur between 6 pm and midnight [7], and 34% of all road fatalities [8]. Alcohol misuse has significant flow-on effects, consuming community, law enforcement and health resources and costing the Australian economy an estimated $14 billion dollars per year [9].

In Australia, night-time alcohol-related violence and disorder have been a significant community concern over the last decade, with numerous high-profile deaths linked with heavy alcohol and other drug use [10]. In response to increasing concerns about public safety in late-night trading areas, the Liquor Amendment Act (LAA) [11] was introduced into law in 2014 in New South Wales (NSW), Australia’s most populous state. The LAA included a range of policy interventions regulating nightlife in defined precincts of NSW, including restrictions on entry and re-entry to venues after 1:30 am (a “lock out” or one-way door policy) and cessation of the service of alcohol from 3:00 am for all licensed premises within the designated area. Other restrictions included a liquor license freeze preventing new venue licenses in the newly established central business district precinct, increased powers for police, restriction on takeaway alcohol sales from 10 pm and harsher sentencing for alcohol-related violence [12]. Alongside the implementation of the LAA, a night-time safe space program designed to support vulnerable patrons was established in three key nightlife areas in Sydney’s inner city, with the aim of reducing alcohol-related harm, violence and crime. This program, the Take Kare Safe Space (TKSS) program, was a key initiative of the plan of management for the Sydney central business district entertainment precinct.

For the purposes of this paper, ‘safe space’ refers to a harm reduction service often run in partnership with health, community, emergency or welfare services to increase public safety and amenity in town centers or other public spaces [13]. Night-time safe spaces have been in operation for several years in many parts of world to reduce the impact of alcohol-related harm, including at music festivals, major events, and in nightlife precincts. Typically, night-time safe spaces provide a combination of medical assessment, first aid, counselling or support, hydration, supervised recovery, and/or practical supports such as the provision of food and directions. In the United Kingdom, a recent study estimated there are up to 45 safe spaces in operation across the country [13]. In Australia, several similar services operate [14,15,16]. The TKSS program was implemented in Sydney in 2014 and was designed to reduce alcohol-related harms, violence and reduce the risk of crime by providing a place where vulnerable young people could access safety and support. Its operations are supported by small teams of paid and volunteer ambassadors who patrol designated precincts to provide alcohol-affected and other vulnerable people in unsafe situations with practical on-the-spot assistance. These teams also manage static safe spaces, providing a place to rest, receive first aid and hydration, charge mobile phones, find transport home, and wait for friends or family. The program is well integrated with other critical safety and health services, including venue security staff, local close circuit-television control rooms, and emergency services including NSW Ambulance and NSW Police, and is a key point of contact for licensed venues dealing with or ejecting heavily intoxicated patrons.

Despite the increasing use of safe spaces as part of local government and community responses to addressing alcohol-related violence, there remains a notable absence of rigorous, independent evaluation of the effectiveness of programs in achieving benefits [17]. To date, only two pilot studies have examined the economic benefit of safe spaces [18,19]. Another study has presented case studies of operation [20], with a third, ongoing, project examining the impact of safe night precincts on a wide range of health and justice outcome measures [21]. Other studies evaluating ‘like’ interventions, including shelter and van programs [16], have shown limited effectiveness on reducing alcohol-related harm and violence, despite high community value and patron use. Similarly, there have been a range of studies into programs and interventions that feature key components of the TKSS service delivery model, including connection with CCTV control rooms and the establishment of radio communication with emergency services and security teams [22,23], but not the full range of their interventions and services. The variance in safe space services makes meaningful comparison of the benefit of programs difficult and highlights the ongoing need for rigorous evaluation of such programs.

Accordingly, this study aimed to determine the impact and return of investment of the TKSS program implemented in Sydney, Australia, as part of a city-wide plan to address alcohol-related violence and disorder.

## 2. Material and Methods

Design. The methodology underpinning the evaluation was guided by the NSW Government’s Program Evaluation Guidelines to examine process, outcome and economic indicators [24]. The evaluation relies on internal program-level data collected by the TKSS ambassadors and qualitative data collected from clients and stakeholders. A mixed-methods approach was embedded into the evaluation framework combining both qualitative and quantitative methods. Process indicators assess uptake of the program; client characteristics including age, gender and intoxication levels; time and duration of contact; services and/or referrals provided. Outcome indicators assess ambassador intervention that seek to prevent harm; and, reductions in demand for acute services (e.g., police incidents, ambulatory care and emergency department (ED) presentations). Economic indicators enable an assessment of whether the economic benefits of the TKSS program outweigh its costs. The time frame of the evaluation spans the period December 2014–April 2019 (inclusive).

TKSS program. The TKSS program commenced operations in Town Hall in the central business district of Sydney, Australia, in December 2014. A second site in Kings Cross commenced operations in July 2015 and a third site at Darling Harbor commenced operation in February 2017. All three sites are in or near the central business district entertainment precinct of Sydney and are renowned for their nightlife. TKSS operated year-round from 10 pm to 4 am on Friday and Saturday nights. Each site of the TKSS program is staffed by groups of 3–4 team members, called ambassadors, including a paid team leader and volunteers. Volunteers are drawn from the public and student recruitment pathways from established relationships between the program and local universities. Volunteer training is provided by the program, and includes instruction on basic first aid, the provision of care for, and communication with, intoxicated patrons, de-escalation and other program safety protocols. Alongside public safety interventions in situations of “risk” (including high-level intoxication and conflict), the TKSS program provides first aid support, escort to accommodation, assistance with accessing transport, a phone charging station, help connecting with friends or family, and general assistance with directions and local information. When the support required exceeds the program’s scope and capacity, ambassadors refer incidents to appropriate emergency or other social services. Integral to the TKSS program is its connection to, and interaction with, other agencies and nightlife service providers to ensure the most appropriate care is provided for those in need of assistance. This includes collaboration with City Rangers, licensed premises, venue security, police, closed circuit-television operators, and transport staff.

As part of program delivery, ambassadors record a range of information relating to client interactions with TKSS, either on a paper-based form or via a mobile phone application. Data recorded included age and gender of the service user, the time support was provided, the length of time each user was in contact with the service, and the perceived level of intoxication of the person/people receiving support (based on the NSW Responsible Service of Alcohol intoxication guidelines [25]). The types of support provided to people receiving assistance is also recorded, including spending time at the Safe Space, Ambassador intervention to de-escalate or avert serious risk of harm, request for directions, phone charging, and receipt of first aid.

Return on investment. Costs were determined by combining information on TKSS operating costs with the value of volunteer time implementing the program. The value of one hour of volunteer time was calculated using Australian Bureau of Statistics data on average weekly earnings and adjusting for a 35 h working week [26]. Benefit was defined as major incidents averted because of ambassador interventions, combined with willingness to pay estimates of the social (community) value attached to averted major road traffic injuries. Given it is uncertain whether an assault, theft or injury would have occurred if the intervention did not happen, ambassadors used their understanding and experience in the city at night to identify those incidents that were of serious risk of occurring.

Table 1 provides an overview of all the interventions provided by Take Kare ambassadors and the subset of those interventions that the ambassadors classified as being at serious risk of occurring. For theft, the category “passed out—valuables visible” is included. For risk of injury, only road-related traffic injuries are considered with a further assumption made that only a fraction of these injuries would be classified as major (derived using the average of major assaults and sexual assaults averted). Over the period December 2014–April 2019 (inclusive), ambassadors’ interventions averted an estimated 735 incidents that were of serious risk of occurring, out of a total of 3633 interventions.

Costs of major incidents averted were derived using estimates of incidents averted together with a cost estimate for each incident. The cost of assault and sexual assault (reported crime) was valued using estimates derived by Byrne et al. (2012) [27]. The cost of theft (reported crime) was valued using estimates by the Australian Institute of Criminology [28]. The cost of a road traffic injury was valued using estimates from the Bureau of Infrastructure, Transport and Regional Economics [29]. All estimates were adjusted to 2018 dollars using consumer price index information from the Australian Bureau of Statistics [30]. Cost per assault was estimated at $18,933; cost per sexual assault at $30,495; cost per theft at $434; and cost per road traffic injury at $5745. A willingness to pay approach was used to estimate the social (community) value of the program [31]. This approach estimates the social cost of death or injury by establishing how much society is willing to pay to reduce the risk (or avoid) fatality or serious injury. This method is preferred to other traditional approaches, such as a human capital approach, as it provides a more representative value of costs to individuals as it takes into consideration other general wellbeing factors, not just earnings and productivity. The value of a statistical life year saved is estimated as $196,484 (in 2018 dollars) and represents the value that society places on preventing one premature death each year [32].

The outcome of the return on investment is reported as a ratio (benefits/costs). Two additional sensitivity analyses were conducted to explore the robustness of results to changes in assumptions. Sensitivity analysis 1 examines changes in the attribution of serious risk of injury diverted from 100% (baseline) to 75%. Sensitivity analysis 2 examines the benefit cost ratio for the year 2016–2017 when the TKSS program was fully operational in three sites—Town Hall, Kings Cross and Darling Harbour.

## 3. Results

User profile. Cumulatively, from December 2014 to April 2019 (inclusive), 66,455 people were supported by the TKSS program. Figure 1 provides an overview of TKSS users by age and gender for those where details were recorded (*n* = 41,395). Sixty-two percent (*n* = 25,460) of users were males and 38% (*n* = 15,859) were female. The largest proportion of service users were aged 18–25 years, accounting for 66% (*n* = 27,279) of the sample, followed by those aged 26–39 (*n* = 7808 or 19%). Of the 8872 assessments completed to determine level of intoxication, 46% (*n* = 4061) were perceived to have had a high level of intoxication, 36% (*n* = 3190) a mild level of intoxication, 11% (*n* = 955) were perceived as sober and 8% (*n* = 666) were perceived to be under the influence of drugs.

Service Use. Sixty-nine percent of all contacts (*n* = 5356) occurred between the hours of 11 pm and 2 am. Seventy percent (*n* = 5392) of users were in contact with the service for between 1 and 20 min, 17% (*n* = 1360) for between 21 and 60 min and 13% (*n* = 981) for greater than one hour. Most users (66% or *n* = 41,396) spent time at the Safe Space, 19% (*n* = 12,645) were supported in other ways (defined as incidents) and 18.7% (*n* = 12,414) requested directions, primarily related to transport.

TKSS program benefits. While it is uncertain whether an assault, theft or injury would have occurred if the intervention did not happen, ambassadors use their judgement based on their understanding and exposure to patterns of behaviour in the city at night. For theft, the category “passed out—valuables visible” is included. For risk of injury, only road-related traffic injuries are considered with a further assumption made that only a fraction of these injuries would be classified as major (derived using the average of major assaults and sexual assaults averted). Table 1 provides an overview of total interventions by ambassadors and those perceived to have de-escalated or averted serious risk of harm. Over the period December 2014–April 2019, serious risk of harm was averted in 735 cases from a total of 3633 interventions.

Return on investment. Table 2 provides an overview of total estimated costs and benefits of the TKSS program over the period December 2014 April 2019. TKSS operating costs were estimated at $2,792,349, including program costs of $1983,198 and the market value of volunteer time at $809,152. Total benefit in terms of costs averted and community value were estimated at $7,461,810. The return on investment is estimated at 2.67, suggesting that a $1 investment results in $2.67 in benefits. The return on investment ratio ranges from a low of 2.00 with a 75% attribution rate to a high of 3.83 when the TKSS was fully operational (i.e., 2016–2017) at three sites. This suggests a range of benefits for every $1 invested of between $2.00 and $3.83.

## 4. Discussion

The purpose of the current study was to determine the impact and return on investment of the TKSS, a harm reduction program embedded within the community to address alcohol-related violence and disorder in the night-time economy. The evaluation relied on an existing framework endorsed by the NSW Government to examine process, outcome and economic indicators [24]. Using a range of data sources, the results of the evaluation demonstrate that the Safe Spaces were extensively utilized, averted harm and resulted in a positive economic investment. These findings are consistent with evaluations of other types of Safe Spaces [13,15,33,34].

The results indicate that the Sydney Safe Spaces were well represented by young men with high levels of intoxication. This is also consistent with global estimates of alcohol use that show higher rates of alcohol use among young men compared to women [35]. Young men are also a group that evidence has consistently show are at high risk of being both perpetrators and victims of alcohol-related violence and harm [36,37,38], and vulnerable to negative biological, neurological, social and psychological effects of alcohol [35]. 

The Take Kare Safe Space aimed to improve public safety and reduce alcohol-related harm and violence by providing a place where vulnerable people could access support in the weekend nighttime economy. One of the objectives of the evaluation was to examine the return on investment. Edmunds et al. (2018) conducted a systematic review of economic evaluations of interventions for high risk young people [39]. Although not specifically focused on Safe Spaces, the authors report a lack of good-quality empirical evidence related to interventions for at risk youth [39]. Their findings were consistent with another review conducted by Knight et al. (2017) that examined the quality and effectiveness of interventions that target multiple risk factors among young people [40] Knight et al. (2017) suggested that more methodologically rigorous evaluations of interventions targeting multiple risk factors among high-risk young people are required, especially for those delivered in community settings [40].

Further, there is a lack of evidence pertaining to community preferences for such interventions. Attaching a value to community benefit is seen as a valuable input in economic evaluations that adopt a social perspective [41]. A recent study by Edmunds et al. (2021) used discrete choice experiments to explore community value and preferences for reducing youth crime and improving community safety using BackTrack [42]. The BackTrack program is a multicomponent community intervention targeting 14–17-year-old high risk young people [42,43,44]. The authors found a strong community preference for youth based programs such as BackTrack that are community based rather than traditional means of dealing with youth crime through punitive measures [42].

The current research estimated a return on investment at 2.67, suggesting that every $1 invested in TKSS resulted in $2.67 in benefits. This return on investment is conservative, as the analysis did not quantify the full spectrum of potential benefits associated with the TKSS program or the value of community benefit. For example, only 20% (*n* = 735) of all interventions made by ambassadors that averted serious risk of harm were included in the formal return on investment analysis out of 3633 actual interventions. Other likely benefits of the program that are difficult to quantify such as improvements in public safety and amenity, more efficient resource allocation for service providers, improved partnership, communication and resourcing, and flow-on effects for tourism and investment, suggest the true return on investment is likely to be much higher. Indeed, the three-month pilot evaluation of the TKSS program estimated a positive return of $9 for every $1 invested, largely due to savings attributed to the cost of police, ambulance and additional medical services [18]. A return on investment analysis of pilot Safe Space initiative in the United Kingdom indicates that for every £1 invested, benefits exceed £1.80 [19]. The authors of the United Kingdom pilot suggest that it is likely that a permanent Safe Space facility has the potential to offer even better value for money, as strong branding and sustained publicity will integrate it into the local night-time economy and lead to higher usage [19]. 

The positive return on investment of the Sydney TKSS is supported by a statistical analysis of routinely collected datasets that showed consistency in the pattern of declining alcohol-related assaults, alcohol-related emergency department presentations and alcohol-related ambulance dispatches within the geographical boundaries of TKSS sites (data not shown but available upon request) since 2009. Although the trends pre- and post-implementation of the LLA and the TKSS program were not significant, the downward trend suggests that Safe Spaces may have had a dampening effect on emergency department presentations and ambulance dispatches (i.e., they keep the demand for ED and ambulance services constant) but not the occurrence of assaults. This would be consistent with previous research on community-based responses to alcohol harms, which showed that alcohol harms can be less in communities with community action responses (compared to communities who do not have those responses), but sustaining reductions in alcohol harms over time is likely to need more targeted and restrictive legislation to control the availability of alcohol, especially at high-risk times [45]. 

Strengths and limitations. This evaluation should be interpreted with reference to the following strengths and limitations. A limitation of the evaluation is the use of internal data collected by Safe Space ambassadors to estimate impacts on alcohol harms. These data were self-reported and collected using an unvalidated tool that may have been prone to recall or reporting bias. Second, during the period of this study (December 2014–April 2019), funding to operate TKSS was sporadic. This sporadic funding impacted on operational capacity to the extent that the Kings Cross Safe Space was closed on Friday nights, operating only on Saturday night. Third, the responsibilities of operating the Safe Spaces transitioned from an external organisation, St John Ambulance, to another organisation—Stay Kind. This transition resulted in additional financial burdens associated with operating the TKSS program. Previously Stay Kind operated from offices provided pro bono with one paid staff member. Bringing the program inhouse required establishing a range of additional systems and processes for running the program including directly employing the team leaders, inhouse management of staff, training and induction of volunteers as well as leasing an office and acquiring vehicles within the constraints of reduced and sporadic funding. Fourth, for the duration of this study, the City of Sydney Central Business District underwent significant capital works that impacted on the mobility of ambassadors and capacity to operate efficiently. This may have also impacted on the number of users to the service. Despite these challenges, the TKSS program has met it objective to provide a harm reduction service where vulnerable young people can access support and a safe place or a safe passage home. A real strength of the program is the ambassadors collection of real-time data. The impacts on alcohol harms are usually not accurately recorded, or captured at all, because they do not come to the attention of relevant authorities [46], or are not captured as part of routine data collection [47]. Ambassadors also demonstrated a willingness to engage with predominantly intoxicated youth and various night time economy participants. Such engagement could also be perceived as a conduit that enabled better partnerships, communication and a more complete suite of resources to manage Sydney nightlife thereby keeping both vulnerable youth and the general public safe. There are likely to be significant flow-on effects of a safer night time environment for tourism and investment.

## 5. Conclusions

Safe Spaces are extensively utilized, particularly by young males with high levels of intoxication, and represent a positive return on investment. Harm reduction programs such as TKSS play a key role in de-escalating conflict and averting the risk of serious harm. Safe Spaces offer a positive return on investment and should play a key role in any city-wide management plan to address alcohol-related violence and disorder in the night-time economy. However, despite the growth of such services, there remains a notable absence of rigorous, independent evaluation regarding the outcomes and/or social benefit of safe space programs. From a policy perspective, there is a need for more high-quality economic evaluations to better inform decisions about competing uses of limited resources. 

## Figures and Tables

**Figure 1 ijerph-18-12111-f001:**
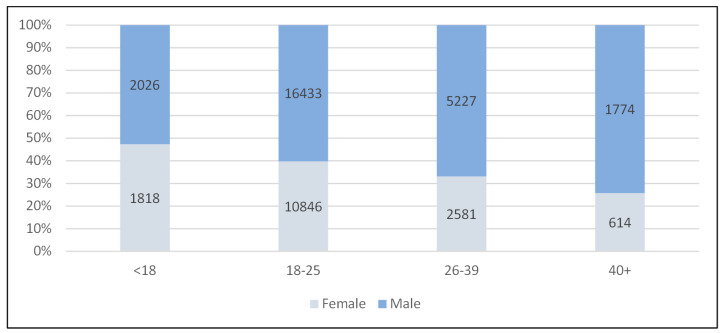
Service users by age and gender.

**Table 1 ijerph-18-12111-t001:** Interventions that averted serious risk of harm (December 2014–April 2019).

Intervention	Ambassador Interventions (14 December–19 April)	Serious Risk of Harm Averted	Proportion of Interventions
Risk of assault minimized	1357	235	17%
Risk of sexual assault minimized	664	50	8%
Risk of theft minimized	904	362	40%
Risk of road traffic accidents minimized	708	88	12%
**Total**	3633	735	20%

**Table 2 ijerph-18-12111-t002:** Return on investment of TKSS program December 2014–April 2019 (inclusive).

	Baseline	Sensitivity Analysis 1 (75% Attribution)	Sensitivity Analysis 2 (Fully Operational TKSS)
Cost			
Cost of TKSS program	$1,983,198	$1,983,198	$470,687
Cost of volunteer time	$809,152	$809,152	$189,168
Total cost	$2,792,349	$2,792,349	$659,855
Benefits			
Cost averted	$6,636,393	$4,977,295	$2,211,956
Community value	$825,417	$619,063	$317,666
Total benefit	$7,461,810	$5,596,358	$2,529,621
Total benefit–total costs	$4,669,461	$2,804,008	$1,869,766
Return on investment	2.67	2.00	3.83

## Data Availability

The datasets generated and analysed during the current study are available from the corresponding author on reasonable request, and subject to approval.

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
