# Peer review of "Impact and Return on Investment of the Take Kare Safe Space Program—A Harm Reduction Strategy Implemented in Sydney, Australia"

_ijerph, 2021, doi:10.3390/ijerph182212111_

Round 1
Reviewer 1 Report
The study need to be revised so that: a) to bring additional information explaining the design of methods used (sect. 2.Materials and methods) to clearly present and b) to improve the sect. Conclusions (limitations of the study, further developments, recommendations for further improvements / extending of the TKSS).
Reviewer 2 Report
Overall, it is a good paper. It has an interesting topic, the results are presented clearly and analysed appropriately, and the findings can be useful for decision-makers. I only have a few observations:
- although the authors mention that the literature on this subject is scarce, they still need a proper literature review to demonstrate adequate understanding of the relevant publications in the field, identify the research gaps, and state their own contributions
- the conclusion section is extremely short and lacks a discussion on the implications of this research for policy making; given the importance of the topic, and the positive return on investment provided by programs such us the one analysed in this manuscript, this section should be enlarged.
- Considering that the program presented in the paper was implemented in Sydney, Australia but the IJERPH journal has an international audience, the authors should try to identify the potential relevance of their research for other regions or countries.
Reviewer 3 Report
1. In the reviewer's opinion, the Discussion section should be expanded more. The Discussion should not mainly refer to the results of the research, but to other works concerning the issue. If not specific, then understood broadly.
2. Conclusions should be corrected. The reviewer suggests setting or signalling a perspective for further research.
3. In general, the article is written correctly.
